# Availability of Ferritin-Bound Iron to Enterobacteriaceae

**DOI:** 10.3390/ijms232113087

**Published:** 2022-10-28

**Authors:** Clemens M. Gehrer, Alexander Hoffmann, Richard Hilbe, Philipp Grubwieser, Anna-Maria Mitterstiller, Heribert Talasz, Ferric C. Fang, Esther G. Meyron-Holtz, Sarah H. Atkinson, Günter Weiss, Manfred Nairz

**Affiliations:** 1Department of Internal Medicine II, Infectious Diseases, Immunology, Rheumatology, Medical University of Innsbruck, 6020 Innsbruck, Austria; 2Christian Doppler Laboratory for Iron Metabolism and Anemia Research, Medical University of Innsbruck, 6020 Innsbruck, Austria; 3Biocenter, Institute of Medical Biochemistry, Medical Universitiy of Innsbruck, 6020 Innsbruck, Austria; 4Department of Laboratory Medicine and Pathology, University of Washington School of Medicine, Seattle, WA 98195-7110, USA; 5Department of Microbiology, University of Washington School of Medicine, Seattle, WA 98195-7735, USA; 6Laboratory of Molecular Nutrition, Faculty of Biotechnology and Food Engineering, Technion-Israel Institute of Technology, Haifa 32000, Israel; 7Kenya Medical Research Institute (KEMRI), Centre for Geographic Medicine Research Coast, KEMRI-Wellcome Trust Research Programme, Kilifi 80108, Kenya; 8Centre for Tropical Medicine and Global Health, Nuffield Department of Medicine, University of Oxford, Oxford OX3 7LG, UK; 9Department of Paediatrics, University of Oxford, Oxford OX3 9DU, UK

**Keywords:** ferritin, *Salmonella* *enterica* subsp. *enterica* serovar Typhimurium, *Escherichia* *coli*, nutritional immunity, superoxide dismutase, reactive oxygen species, iron metabolism, siderophore

## Abstract

The sequestration of iron in case of infection, termed nutritional immunity, is an established strategy of host defense. However, the interaction between pathogens and the mammalian iron storage protein ferritin is hitherto not completely understood. To better characterize the function of ferritin in Gram-negative infections, we incubated iron-starved cultures of *Salmonella* Typhimurium and knockout mutant strains defective for major iron uptake pathways or *Escherichia coli* with horse spleen ferritin or ionic iron as the sole iron source. Additionally, we added bovine superoxide dismutase and protease inhibitors to the growth medium to assess the effect of superoxide and bacterial proteases, respectively, on *Salmonella* proliferation and reductive iron release. Compared to free ionic iron, ferritin-bound iron was less available to *Salmonella*, but was still sufficient to significantly enhance the growth of the bacteria. In the absence of various iron acquisition genes, the availability of ferritin iron further decreased. Supplementation with superoxide dismutase significantly reduced the growth of the Δ*entC* knockout strain with holoferritin as the sole iron source in comparison with ionic ferrous iron. In contrast, this difference was not observed in the wildtype strain, suggesting that superoxide dismutase undermines bacterial iron uptake from ferritin by siderophore-independent mechanisms. Ferritin seems to diminish iron availability for bacteria in comparison to ionic iron, and its iron sequestering effect could possibly be enhanced by host superoxide dismutase activity.

## 1. Introduction

Non-typhoidal *Salmonella enterica* serovars and pathogenic types of *Escherichia coli* are two of the major causes of enterocolitis worldwide, with some strains even being considered as causes of emerging diseases [1,2]. These and other pathogens need iron to achieve their full virulence [3]. In the case of infection, the host can take advantage of bacterial iron demand by depriving pathogens of this essential nutrient. This strategy is termed nutritional immunity and has been investigated in many infection studies [4,5,6,7]. However, many basic mechanisms are hitherto incompletely understood. One of these is the interaction between the host protein ferritin and bacteria.

Ferritin is mainly known as the iron storage protein and for its role in protecting cells from iron-mediated damage by reactive oxygen species (ROS) [8,9]. Besides this, it acts as an acute phase protein and has immune-modulatory functions [10,11,12]. In line with this, systemic *Salmonella* infection results in increased ferritin levels in serum and tissues [13,14]. On the other hand, ferritin may serve as a possible bacterial iron source [15,16]. This idea is corroborated by the fact that elevated serum ferritin and tissue iron overload are associated with Gram-negative infections in patients with thalassemia [17,18,19]. In these patients, iron may accumulate because of increased dietary iron absorption, enhanced yet ineffective hemoglobin synthesis, and/or blood transfusions.

Mammalian ferritin is a 24-meric protein, approximately 480 kDa in size, and it consists of two of types subunits: An H-subunit, containing a ferroxidase active site, and an L-subunit, which is primarily involved in iron nucleation within the ferritin cavity and contributes to ferritin stability [20]. Ferritin’s iron storage capability is accompanied by its role as an ROS defense protein. In this regard, ferritin can use molecular oxygen and ROS, such as hydrogen peroxide, to oxidize ferrous to ferric iron [21]. This ferric iron subsequently makes up the core of ferritin in the form of ferrihydrite and hematite with traces of magnetite and maghemite [22]. The oxidation of iron is facilitated catalytically by either the ferroxidase active site of the H-subunit or the mineral core itself [23]. As a result, iron is stored in a relatively safe form, which does not substantially contribute to ROS production via Fenton chemistry. Ferritin’s importance in that manner is further emphasized by its high conservation through all forms of life, and is supported by the finding of embryonic lethality of mice in case of a complete H ferritin gene knockout [24,25]. Additionally, a complete L-ferritin gene knockout is also embryonically lethal in mice, if there is only one functional H-ferritin allele [24].

*Salmonella enterica* subsp. *enterica* serovar Typhimurium ATCC 14028 (*S.*Tm), which is a pathogenic non-typhoidal *Salmonella* strain, and *Escherichia coli* O18K1 (*E. coli*) have a wide array of iron acquisition mechanisms including enterobactin, SitABCD and the Feo system, considered as their three most important ones.

Enterobactin and its glycosylated derivatives, the salmochelins, are high affinity catechol siderophores, which coordinate ferric iron as a hexadentate ligand. Due to enterobactin’s extremely high affinity for ferric iron (K = 10^52^ M^−1^), enterobactin is seen as an affinity sink, which competes very effectively for ferric iron with host proteins, like transferrin (K = 10^20^ M^−1^) [26,27,28]. Once iron is bound, ferric enterobactin is actively taken up into the periplasmic space via FepA, a TonB-dependent transporter [29]. Subsequently, it enters the cytoplasm through the ABC-transporter FepBCDG and is hydrolyzed by the serine hydrolase Fes. Thereby, the ferric iron is reduced, enabling its release from enterobactin in the process [30,31,32,33]. Salmochelins are taken up and processed in a similar yet distinct fashion via products of the *iroA* gene cluster [34,35]. Enterobactin has also been shown to have a function in ROS defense, which is distinct from its ability to chelate iron, [36,37].

SitABCD is an ABC-transporter, which is mainly acknowledged for Mn^2+^ uptake [38]. However, it has been shown that the affinity can change in favor of iron, depending on the activities of other transporters for manganese and iron [39]. Therefore, SitABCD may take up iron into the cytoplasm from the periplasmic space in an energy-dependent manner [40].

In contrast, the Feo system is regarded to be exclusively involved in iron uptake [41]. Functionally, FeoB is an energy-dependent ion channel, located in the cytoplasmic membrane [41]. FeoA is involved in the iron transport of FeoB and in some bacteria even necessary for iron uptake via FeoB, but its precise function is hitherto unknown [41,42,43]. FeoC has an oxygen-sensing site, preventing degradation of FeoB in an anaerobic environment. In contrast, in an oxygen-rich environment, FeoC and consequently FeoB are quickly degraded. Hence, the Feo system seems to play a major role primarily in anaerobic environments [44].

In the case of infection, the host and the pathogen both strive for iron. Besides hemoglobin, ferritin is one of the most abundant iron containing proteins. Thus, ferritin is assumed to be a prime target for bacterial iron acquisition and it has been shown previously in *Pseudomonas aeruginosa* and *Burkholderia cenocepacia* to be a viable iron source [15,16].

The use of host ferritin as a bacterial iron source is facilitated by proteolytic degradation, and 4-(2-Aminoethyl)benzenesulfonyl fluoride (AEBSF), a water-soluble, irreversible serine protease inhibitor, is able to prevent ferritin degradation [16].

An additional mechanism to acquire iron from ferritin is reductive dissolution of its ferrihydrite core, especially in the interaction with proteases [15]. *S.*Tm has extracellular iron reducing activity as well, although this has not been further characterized as yet [45]. As extracellular reductases of other Enterobacteriaceae, such as those expressed by *E. coli*, require flavins as cofactors, flavoenzymes may be important for *S.*Tm as well [45]. However, the role of flavoenzymes as primary reductants in an aerobic environment, such as a host cell, are unclear because of the highly variable reactivity with oxygen. Thus, some flavoenzymes react very quickly with dioxygen (O_2_), forming ROS, such as hydrogen peroxide (H_2_O_2_) or superoxide (O_2_^−^) in the process, while others are resilient to oxidation by dioxygen [46,47]. Therefore, we investigated the effect of O_2_^−^, which is able to reduce ferric to ferrous iron.

Inside the host cell, the superoxide is constantly generated in the mitochondria via one electron transfer to O_2_ during aerobic respiration, but it can also be found in the cytoplasm [48,49,50]. To prevent cell damage from the constant generation of O_2_^−^, an array of different enzymes, called superoxide dismutases (SOD), are used. These catalyze the reaction 2 H^+^ + 2 O_2_^−^ → H_2_O_2_ + O_2_. Thereafter, hydrogen peroxide is further detoxified by a wide array of enzymes and molecules with antioxidant activity, but may also be used as a substrate for ferritin’s iron core formation [21,51].

Superoxide can also be generated during non-enzymatic glycation reactions between reducing sugars and amino acid groups in the early stages of the Maillard-reaction [52]. This is of relevance, because the superoxide is able to reductively mobilize iron from ferritin, as several studies have demonstrated [53,54,55,56]. Additionally, ferritin-bound iron can be mobilized by strong iron(III)-ligands with superoxide as a catalyst [57]. Due to these mechanisms, the conversion of superoxide to hydrogen peroxide by SOD in host cells may impair the access of bacteria to ferritin-stored iron by reductive and/or siderophore-dependent pathways. Data about a putative synergy between ferritin and SOD in nutritional immunity to bacterial infections remains scarce. We therefore also studied the effects of mammalian SOD on the growth of *S*.Tm in the presence of holoferritin.

Although the role of ferritin as an acute-phase reactant is well established, little is known about its function in regard to iron sequestration from Gram-negative pathogens. We found that ferritin-bound iron is available for *S.*Tm, predominantly by an enterobactin/salmochelin-dependent pathway. Opposingly, host SOD may mitigate reductive iron acquisition from ferritin. Our data provide new insights into the role of ferritin in host-pathogen interaction and highlight the importance of synergies of host proteins in the context of nutritional immunity.

## 2. Results

### 2.1. Ferritin-Bound Iron Is Less Available for Enterobacteriaceae Than Ionic Ferrous Iron

We first assessed the growth of *S.*Tm and *E*. *coli* in the presence of increasing concentrations of ferrous iron or equimolar amounts of ferritin-bound iron. The addition of either form of iron promoted the growth of *S.*Tm and *E. coli* in comparison to the iron-free growth medium IMDM alone. Yet, at concentrations as low as 0.233 (*S.*Tm) or 1.6 µM (*S.*Tm and *E. coli*), ferrous iron resulted in a significantly higher growth-promoting effect. This difference disappeared with rising concentrations of iron (Figure 1a,b). However, the supplementation with apoferritin also resulted in significantly enhanced bacterial growth in comparison to IMDM (Figure 1c,d). Hence, we subsequently normalized the growth data of the highest iron groups, grown in 50 µM of ferrous iron or ferritin-bound iron, to the iron free group or apoferritin with an equimolar protein concentration as holoferritin, respectively. We deemed this normalization necessary to separate the growth-promoting effect of ferritin-bound iron from the one of the almost iron-less protein alone. Again, a significant difference between bacteria treated with free iron or holoferritin was observed (Figure 1e,f). The growth-promoting effect of apoferritin on *S.*Tm was completely abolished by the addition of a protease inhibitor cocktail (PIC), indicating that apoferritin mainly served as a protein source for bacteria (Figure 1g).

### 2.2. Iron Acquisition from Ferritin Is Still Possible in Absence of Enterobactin and Salmochelins

Next, we investigated the importance of the different *Salmonella* iron uptake systems for iron acquisition from holoferritin, employing different mutants with deletions of central iron acquisition pathways. The iron starved knockout mutant strains for enterobactin as well as salmochelins (Δ*entC*) and the one with combined *entC*, *sitABCD* and *feo* deletion (Δtriple) were severely impaired in growth when no iron was supplemented (Figure 2a). Iron is an important co-factor in many ROS detoxifying enzymes, and enterobactin, missing in both the Δ*entC* and Δtriple strains, also acts as a radical scavenger [36,37,58,59,60]. We therefore assumed that our observations may be partly attributable to an impaired defense against ROS. Indeed, the addition of n-acetylcysteine (NAC), a known antioxidant, completely eliminated the growth difference in the iron-deprived strains (Figure 2b). Furthermore, in iron-supplemented medium, the knockout mutants showed a much larger increase in bacterial growth than the wildtype (WT) (5.774 ± 0.3844%, 17.46 ± 0.3912% and 18.51 ± 0.3844%; (mean difference in fold change of logarithmized data ± SEM) increase for the WT, Δ*entC*, Δtriple, respectively). However, we still observed a small but significant difference between the wildtype and the knockout strains (Figure 2c). The observation that the knockout strains not only compensate their growth deficiency under iron depriving conditions but nearly match the growth of the WT when iron is available is consistent with a role of iron in enzymes for ROS defense. Otherwise, the increase in bacterial growth in the knockout mutants would be expected to be similar to the increase in the WT. Ferrous and ferric iron as well as holoferritin at an iron concentration of 5 µM did not cause any significant difference in growth in the WT. Opposingly, the same iron sources resulted in significantly reduced growth in both knockout mutant strains in comparison to the WT (Figure 2c–e). Thereby, ferrous iron supplementation resulted in no significant difference in growth between the Δ*entC* and Δtriple mutant (Figure 2c). In contrast, the Δtriple mutant showed a significantly lower growth-promoting effect than the Δ*entC* mutant, when ferric iron was used instead (Figure 2d). However, the most prominent effect on growth became evident when holoferritin was added as an iron source. There, we observed a significant reduction in bacterial growth, which increased with the number of deleted iron uptake systems (Figure 2e). These findings indicate that not only the oxidation state of ferritin-bound iron [22] affects the availability to the knockout strains, but also other factors. We suggest that some of these factors may include the presence of iron in aggregates [22,61] or direct effects of the protein shell. However, the mutant strains still benefitted from ferritin as an iron source, indicating that free iron, released from ferritin, is present in the medium. Therefore, we used atomic absorption spectrometry (AAS) to estimate the release of iron from ferritin into the medium in our experimental setup without the assistance of bacteria and found it to be 0.147 ± 0.071 µM (mean ± SD; N = 15, n = 5) after the removal of ferritin-bound iron via ultrafiltration. In comparison, no iron was detected in this lot of the medium alone. We suggest that one of the mechanisms contributing to this iron release is the formation of O_2_^-^ in the medium through a glycation reaction of the reducing sugars and amino acids present in the medium [52].

### 2.3. The Serine Protease Inhibitor AEBSF Impairs Iron Utilization in the WT but Not the ΔentC Mutant

Next, we reasoned that *Salmonella* may require a bacterial protease for the acquisition of ferritin-bound iron. To test this idea and to find the type of protease supposedly facilitating iron acquisition from ferritin, we added a PIC (Figure 3a) or the individual protease inhibitors to the cultures (Figure 3b). This experiment revealed that AEBSF alone had the same growth-inhibiting effect as did the PIC (Figure 3b). Unexpectedly, we also observed the same growth-inhibiting effect of the PIC in the group supplemented with ionic ferrous iron (Figure 3a). This observation led us to hypothesize that AEBSF impairs iron assimilation via blockage of the serine hydrolases Fes and IroD, which are essential for the release of iron from enterobactin and salmochelins, respectively. Therefore, AEBSF was tested in both the wildtype strain and the Δ*entC* strain, which should not be affected by this mechanism. In order to ameliorate the effect of ROS on the Δ*entC* strain, observed in ambient air, these experiments were conducted in an environment resembling the microenvironmental tissue oxygen concentration of the host under physiologic conditions more closely (5% O_2_) [62]. The change in environment alleviated the difference between the WT and the Δ*entC*, while the residual difference still remained significant (−13.12 ± 0.4404%, *p* < 0.0001, and −3.230 ± 0.3384%, *p* < 0.0001, for ambient air and 5% O_2_, 5% CO_2_, respectively; mean fold difference of logarithmized growth data ± SEM) (Appendix A). AEBSF alone resulted in a significant iron-independent reduction of growth in the WT strain (Figure 3c). However, the effect was more pronounced in the Δ*entC* strain (Figure 3d). Therefore, we normalized the data to their corresponding control to determine the sole growth-promoting effect of iron without the iron-independent effect of AEBSF (Figure 3d,f). AEBSF abrogated the relative growth-enhancing effect of ferrous iron and ferritin-bound iron selectively in the WT. In contrast, AEBSF lacked its effect on iron-dependent growth in the Δ*entC* strain (Figure 3e,f). We interpreted this finding as an impaired iron assimilation from enterobactin and salmochelins with subsequent trapping of iron inside of them, as the serine hydrolases Fes and IroD are necessary for iron release from enterobactin and salmochelins, respectively, and may be blocked by AEBSF [33].

### 2.4. Superoxide Is Necessary for Bacterial Iron Acquisition from Ferritin in the Absence of Enterobactin and Salmochelins

Superoxide influences iron storage in ferritin. To see whether the superoxide anion also affects bacterial iron acquisition, we added purified SOD to the medium in an atmosphere of 5% O_2_ in order to mimic the host tissue microenvironment more closely [62]. In these experiments, we saw that the addition of SOD had no significant effect on the growth of the WT strain with or without ferrous iron or holoferritin (Figure 4a). As the catalytic reaction is speculated to only need a small amount of superoxide to remove iron from ferritin effectively, we also tested an excess amount of SOD with the same result (Appendix A). Next, we evaluated the effect on the Δ*entC* mutant, which is supposed to be reliant on reductive mechanisms. In the Δ*entC* mutant, the addition of SOD resulted in a significant reduction of bacterial growth both in medium (3.568 ± 0.3433%; mean difference in fold change of logarithmized data ± SEM, *p* < 0.0001) and in the presence of holoferritin (0.9872 ± 0.3433%; mean difference in fold change of logarithmized data ± SEM, *p* = 0.0163) (Figure 4b). However, *S.*Tm needs only a tiny amount of iron to grow (Figure 1a), and even an excess of SOD did not result in a difference in growth in the WT (Appendix A). We therefore suspected that the restrictive effect of SOD on bacterial iron acquisition and/or metabolism may be too small to be detected via bacterial counts alone. Hence, we used qPCR as a highly sensitive method to evaluate the expression of *ryhB1*. RyhB1 is one of two orthologous small non-coding RNAs (sncRNA) which are strongly regulated by iron [63]. Thereby, iron starvation results in an upregulation, while iron sufficiency suppresses the expression of *ryhB1*. The expression data showed no difference in the expression of *ryhB1* in the WT strain in response to SOD (0.1002 ± 0.06651 mean fold difference ± SEM, *p* = 0.2578) (Figure 4c), while in the Δ*entC* mutant, SOD and ferritin treatment led to significantly higher *ryhB1* expression levels compared to ferritin treatment alone (−0.2207 ± 0.08938 mean fold difference ± SEM, *p* = 0.0483) (Figure 4d). Furthermore, holoferritin treatment resulted in considerably less downregulation of *ryhb1* than the supplementation with an equimolar amount of ionic ferrous iron. In addition to that, we observed that the Δ*entC* mutant showed significantly higher *ryhB1* expression than the WT when holoferritin was used as a sole iron source (Appendix A). These findings corroborate our previous findings that ferritin-bound iron is less available than ionic ferrous iron and that the Δ*entC* mutant is impaired in its ability to acquire iron from ferritin. As Fur and subsequently *rhyB1* are also influenced by oxidative stress, we assessed the expression of Alkyl hydroperoxide reductase C (AhpC), which is a direct target of peroxide sensing master regulator OxyR and does not need iron as a co-factor. We observed no significant difference in *ahpC* expression between the groups with or without SOD (−0.07108 ± 0.1102%, *p* = 0.7708, and −0.03273 ± 0.1126%, *p* = 0.7722 in the ionic iron and holoferritin group, respectively; mean fold difference ± SEM), indicating that the effect is indeed due to less availability of iron (Figure 4e,f).

## 3. Discussion

Our data provide the first evidence that ferritin is a feasible iron source for *S.*Tm and *E. coli* in liquid cultures in vitro, which is in line with observations made in vivo [64]. However, our study also demonstrates that ferritin-bound iron is less available for bacteria than ionic iron on an equimolar basis. This indicates that the iron sequestration by ferritin not only prevents Fenton chemistry, but also constitutes an obstacle for bacterial iron acquisition by these two relevant human pathogens. Mechanistically, the oxidation state of iron, its presence in the form of mineral chunks inside the protein shell of ferritin and the protein fraction of holoferritin are probably responsible for this effect. In comparison to ferritin-bound iron, dissolved ferric iron resulted in enhanced growth while being less obtainable than ferrous iron for knockout mutants lacking enterobactin and salmochelins. Moreover, previous studies have shown an inhibition of proteases to mitigate iron acquisition from ferritin [15,16].

Based on our data, we suggest that *Salmonella* relies on two main mechanisms of bacterial iron acquisition from ferritin. Firstly, siderophores may constitute the most important contributors to iron assimilation from ferritin in vitro. Specifically, approximately half of the growth-promoting effect of free iron supplementation is lost when ferritin is added to the Δ*entC* mutant instead. In addition, the other half may be taken up as ferrous iron via SitABCD, because *S.*Tm ATCC 14028 is not able to accumulate the FeoB channel in an aerobic environment. This is because of proteolytic degradation of the oxygen-sensing protein FeoC and subsequently FeoB [44]. This is also in agreement with an in vivo study in BALB/c mice, demonstrating that a *feoB* knockout strain did not result in an attenuation of systemic disease [65]. In contrast, another study showed FeoB to be necessary for full virulence of a different *S.*Tm isolate in 129/Sv *Nramp1^−/−^* mice. Nevertheless, in that study, a *sitABCD* knockout was less virulent than the *feo* knockout strain [66]. However, in our experiments, the triple knockout, missing *sitABCD*, still benefited from ferritin as an iron source, suggesting that the manganese transporter MntH, which is also known to transport ferrous iron, may play a role as well in this experimental setup [67]. Altogether, full iron acquisition from ferritin seems to be dependent on both siderophores and reductive mechanisms.

Another discussed topic is the facilitation of iron release from ferritin by pathogen-derived proteases [15,16]. However, in our experiments, the protease inhibitor AEBSF abolished the iron-dependent growth-promoting effect in the WT when a protein-free iron source was used. Therefore, any effect of the protease inhibitor cocktail used on iron assimilation from ferritin by *S.*Tm remains unresolved. Rather, we attributed the effect of AEBSF mainly to the blockage of the serine hydrolases, which is necessary for iron release from ferric enterobactin as well as salmochelin, and the subsequent iron trapping inside these siderophores [33]. We reasoned that this was because in the presence of AEBSF, the Δ*entC* mutant showed similar iron-dependent growth as when AEBSF was absent in bacterial cultures. However, this iron-trapping effect is likely only relevant in bacteria relying on hydrolyzation of siderophores via serine esterases to release iron, because *Pseudomonas aeruginosa* only showed reduced growth upon the addition of a PIC selectively in the holoferritin-supplemented group, and not when a protein-free iron source was used [15]. This is likely to be due to *Pseudomonas* using reductive mechanisms to free iron from its siderophores [68]. Furthermore, this finding suggests that proteases facilitate iron acquisition from ferritin in at least some bacteria. In order to show the effect of the protease in enterobactin-utilizing bacteria, a knockout mutant would be needed, but due to the responsible protease being unknown and because of redundant functions of proteases, this task is difficult to accomplish. However, a possible candidate could be the Lon protease, which has been shown recently to be secreted massively under iron starving conditions by another member of the Enterobacteriaceae, *Klebsiella pneumoniae* [69]. Moreover, bacterioferritin, which is a bacterial iron storage protein related to ferritin, has been identified as a target of the Lon protease in *E. coli* [70]. In some *Salmonella* sp., extracellular protease activity was found. However, only a low number of the responsible proteases were identified, such as the membrane-bound PgtE of *Salmonella* Typhimurium [71,72,73]. Another possible candidate could be a hitherto unknown serine protease autotransporter of Enterobacteriaceae, SPATE [73].

As reductive mechanisms of iron release from ferritin seem to play an important role for iron acquisition from ferritin, we investigated the superoxide anion. Superoxide plays a very important role in host defense as it is a product of NADPH oxidase during the oxidative burst in professional immune cells [74]. However, as it is also able to mobilize iron from ferritin, we investigated its putative role in bacterial iron acquisition. To this end, we added purified SOD and studied its effects on the growth of bacteria in the presence of ferritin-bound iron. The observation that the addition of SOD made no difference in bacterial growth or the expression of *ryhB1* in the WT strain, in contrast to the Δ*entC* mutant, indicates that enterobactin and supposedly salmochelins liberate iron from ferritin by direct chelation, possibly by entering the ferritin shell. This would also be in agreement with previous findings that enterobactin passes the outer bacterial membrane through the narrow tunnel of FepA with an approximate diameter of 3 Å, which is smaller than the approximate inner diameter of 3–5 Å of the three- and four-fold channels of ferritin [75]. Additionally, the data of the Δ*entC* mutant show that iron is mobilized in a biologically relevant amount by superoxide in this setup, and the inhibition of this mechanism prevents *S.*Tm from utilizing this freed iron. This finding indicates a possible involvement of SOD in nutritional immunity by facilitating a shift of free iron to the inside of ferritin. In other words, SOD supports ferritin’s function in iron sequestration from microbes. This synergistic function of ferritin and SOD in nutritional immunity, however, may only be relevant in undermining the reductive mechanisms of bacterial iron acquisition from ferritin.

As WT *S.*Tm was able to circumvent the shift of iron inside of ferritin by SOD via their catecholate siderophores, the relevance of this mechanism must be the topic of further research in infection models. However, the sequestration of enterobactin via the host defense protein lipocalin-2 might possibly result in a similar growth-reducing effect as that seen in the Δ*entC* mutant [76,77]. However, we still found a growth-promoting effect of ferritin for the Δ*entC* mutant when SOD was added. These data suggest that *S.*Tm has other ferrireductases capable of mobilizing ferritin-bound iron, consistent with previous findings [45]. Furthermore, *S.*Tm actively diminish the transcription of SOD together with iron-sequestering proteins like ferritin and ferroportin by interfering with the upstream regulator Nrf-2 via the SPI-2 effector protein SpvB [78,79,80]. SpvB down-regulates Nrf-2 and additionally marks it for proteasomal degradation [79]. Nrf-2 has previously been shown to be an important regulator in nutritional immunity, depriving intracellular pathogens of iron by enhancing the egress of iron via ferroportin [80]. Furthermore, Nrf-2 activation upon infection has also been observed in non-professional immune cells, emphasizing the importance of this mechanism [81,82].

In mammals, ferritin is present both in the extracellular space and in intracellular compartments. The growth of bacteria in liquid media mimics extracellular conditions, but ferritin is also of pivotal importance for iron-sequestration in the intracellular space [64]. Macrophages and other cell types keep levels of labile iron in their cytosol low and incorporate iron that is not required for metabolic needs into ferritin. This is especially relevant because several members of the Enterobacteriaceae such as *Klebsiella pneumoniae*, *Yersinia enterocolitica* and *S*.Tm, as used in our study, have a facultatively intracellular life-style. These bacteria may cause invasive infections in human subjects with iron overload secondary to thalassemias [17,18,19]. Of note, thalassemia syndromes, other hemoglobinopathies and repetitive blood transfusions result in iron overload of monocytes, macrophages and other cell types and in elevated ferritin levels in serum and tissues [4,5,6,7]. Therefore, inside macrophages, ferritin plays a central role in host-pathogen interaction and iron-mediated nutritional immunity against intracellular microbes. On the one hand, macrophages incorporate iron into ferritin for sequestration from microbes. On the other hand, facultatively intracellular microbes may trigger autophagy to access ferritin-stored iron. This idea is supported by a recent study demonstrating ferritin to be compartmentalized in the cytosol as a liquid phase condensate with NCOA4 [83]. This may alter the way previous and present studies have to be interpreted, as the availability of ferritin-bound iron present in this condensate has not been studied thoroughly. In that matter, one study investigating the infection of bladder epithelial cells with an uropathogenic *E. coli* (*UPEC*) strain is of special interest [84]. *UPEC* were found to follow ferritin-bound iron into autophagosomes, and a knockdown of NCOA4 reduced bacterial growth, while an autophagy knockdown seemingly reduced bacterial growth even further [84]. These findings indicate that ferritin dispersed in the cytosol might possibly be more available to *UPEC* than that inside the liquid phase condensate. However, ferritin-bound iron may become available to *UPEC* following iron release via ferritinophagy [84]. In addition, changes in the expression of other iron-handling proteins and in the lysosomal pH may also be relevant [82,85,86,87]. Moreover, in our study we found that *E. coli* O18K1, which is also a uropathogenic *E. coli* isolate, can utilize ferritin-bound iron. Likewise, the access of *Salmonella* to ferritin could be similar, as *Salmonella* is compartmentalized in the *Salmonella*-containing vacuole (SCV), and during the later stages of infection, *Salmonella* can access endosomal cargo via the *Salmonella*-induced filaments [88]. Furthermore, *Salmonella* might be able to encounter ferritin just after invasion, as it repairs the SCV, which was damaged by SPI-I, with the host’s autophagy machinery, possibly by autophagosome fusion with the SCV [89]. Thereby, another possible mechanism by which SOD may act synergistically with ferritin is by reducing autophagy, which has been demonstrated to be positively regulated by O_2_^−^ [90]. It is thus tempting to speculate that ferritin and SOD cooperate in withdrawing iron from Enterobacteriaceae and other human pathogens.

## 4. Materials and Methods

### 4.1. Reagents, Media and Laboratory Supplies

All reagents, media supplements and laboratory supplies were bought from the corporations Sigma Aldrich (St. Louis, MO, USA), Thermo Fisher Scientific (Waltham, MA, USA), Merck Millipore (Burlington, MA, USA), and Corning (Corning, NY, USA). Holo- and apoferritin from equine spleen (F4503 and A3641, respectively) were acquired from Sigma Aldrich. IMDM iron content was measured via AAS and was not detectable, and 4.46 nM in the lots used for the experiments depicted in Figure 1, Figure 2, Figure 3 and Figure 4, respectively. The ferritin solutions were analyzed by the manufacturer and estimated to contain 10.4 mg/mL of iron (186.23 mM solution) as well as 61 mg/mL and 54 mg/mL of protein for the holoferritin and apoferritin solution, respectively. This corresponds to a mean iron loading of ferritin of approximately 1800 Fe atoms per shell. We determined the iron content of the apoferritin solution via AAS to be 44.6 µM. The ionic iron sources ferrous sulfate heptahydrate from Fluka (44970) or ferric sulfate hydrate from Sigma Aldrich (F0638) were used. N-acetylcysteine was acquired from Sigma Aldrich (A9165). After dissolution, NAC was stored in an N_2_ atmosphere to protect it from oxidation. The protease inhibitor cocktail (P8340) and 4-(2-Aminoethyl)benzenesulfonylfluorid hydrocholoride as well as E-64 were purchased from Sigma Aldrich (A8456 and E3132, respectively). Aprotinin (A1153), Leupeptin (L2884) and Pepstatin A (P5318) were obtained from Sigma Aldrich. SOD (Mn and Cu/Zn—SOD from bovine erythrocytes) was purchased from Sigma Aldrich (S7571). We determined the iron content of the SOD stock solution via AAS to be between 0.46–0.69 µM.

### 4.2. Bacterial Strains and General Growth Conditions

All experiments were conducted using *Escherichia coli* O18K1 and *Salmonella enterica* subsp*. enterica* serotype Typhimurium ATCC 14028, from which derivative knockout mutant strains were used. These are *S.*Tm MLC774 (Δ*entC::aph*, Δ*sit::bla* and Δ*feo::Tn10*) and MLC619 (Δ*entC::aph*) [91,92]. If appropriate, stocks were selected with kanamycin, ampicillin and tetracycline before freezing at concentrations of 50 µg/mL, 100 µg/mL and 10 µg/mL, respectively. For all experiments, iron starved viable cultures of the respective strains were used. The iron starvation was accomplished by inoculating 10 mL of Iscove’s Modified Dulbecco’s Medium (IMDM) with glycerol stocks of viable cultures in lysogeny broth (LB) and letting it grow over night in an orbital shaker at 37 °C and 200RPM. Subsequent viable cultures were made by inoculating 200 µL of the overnight culture into fresh 10 mL of IMDM at the same conditions. We defined a viable culture as having an optical density at 600 nm (OD_600_) of 0.5–0.6. This corresponds to the bacteria being in their mid-logarithmic growth phase. When the culture reached the desired optical density it was put on crushed ice and counted with a CASY^®^ cell counter and analyzer (OLS, OMNI Life Science, Bremen, Germany).

### 4.3. General Experimental Conditions

All experiments were conducted using IMDM and adding supplements to it. All stimulations were prepared as mastermixes and then split into 3 vessels. Each vessel was then inoculated with 2 × 10^6^ viable bacteria per milliliter as quantified by the viable count of the cell counter. The cell counter determines viability by measuring the electrical resistance of intact bacterial membranes. A loss in electrical resistance, due to permeated membranes, is thereby regarded as dead cells. Experimental cultures were grown for approximately 4.5 h and then put on ice. Afterwards, the viable count/mL of each replicate was assessed with the CASY^®^ cell counter.

### 4.4. Specific Experimental Conditions

The experiments to assess whether bacteria are able to use ferritin as an iron source were all conducted in round bottom tubes with a filling volume of 2 mL of medium or mastermix. Furthermore, they were conducted in ambient air in a GFL 3031 orbital shaker at 37 °C and 200 RPM. In the experiments assessing the effect of ROS on the mutant strains via the addition of n-acetylcysteine, the medium was treated as follows to account for the enhanced acidity due to the addition of NAC: IMDM was shaken overnight in an acid washed Erlenmeyer flask to get a pH stable solution by evaporating present bicarbonate. Afterwards, the medium was titrated to a pH of 7.4 with 1 M HCl or NAC and 1 M HCl for the control and the experimental group, respectively. The experiments assessing the effect of SOD and AEBSF were undertaken in 24-well plates with a filling volume of 1ml of medium or mastermix. Additionally, these experiments were carried out in physiologic tissue oxygen levels on a Heidolph Rotamax 120 orbital shaker at 200 RPM inside a Memmert ICO105med incubator, set to result in ambient conditions of 5% O_2_, 5% CO_2_ and 37 °C [62]. Empty wells of the plates were filled with ultrapure water to mitigate the evaporation of bacterial cultures, which was overall similar between groups and replicates. As visible aggregates formed in the experiments using 24-well plates, cultures were resuspended until cultures were homogenous with a 1 mL pipette before counting.

### 4.5. Quantification of Iron Released from Ferritin

To examine the amount of iron released from the holoferritin solution used in our experimental setup, we diluted the holoferritin stock solution the same way as used in our liquid culture experiments supplemented with 5 µM of ferritin-bound iron. We then incubated 2 mL of the resulting ferritin-supplemented IMDM solution in round bottom tubes at 37 °C for 4.5 h in ambient air. After that, 500 µL of each solution was filtered through Amicon^®^ Ultra 0.5 mL Centrifugal Filters with a molecular weight cutoff of 10 kDa in order to restrain the protein-bound iron in the filter. The mean iron concentration of the filtrates was 0.147 ± 0.071 µM (mean ± SD, n = 5). Iron was quantified by graphite furnace atomic absorption spectrometry (M6 Zeeman GFAA-Spectrometer; Thermo Scientific) at 248.3 nm and Zeeman background correction using 1100 °C ash temperature and 2100 °C atomization temperature under argon atmosphere.

### 4.6. Quantitative Real-Time PCR

The quantitative real-time PCR was carried out as described elsewhere [14]. In brief, total RNA isolation was prepared using acid guanidinium thiocyanate-phenolchloroform extraction with peqGOLD Tri-Fast™ (Peqlab). For reverse transcription 0.3 μg RNA, random hexamer primers (200 ng/μL) (Roche), dNTPs (10 mM) (GE Healthcare LifeSciences) 20 U RNasin (Promega) and 200 U M-MLV reverse transcriptase (Invitrogen) in first-strand buffer (Invitrogen) were used. Ssofast Probes Supermix and Ssofast EvaGreen Supermix (Bio-Rad Laboratories GmbH) were used according to the manufacturer’s instructions. Real-time PCR reactions were performed on QuantStudio 3 and 5 real-time PCR systems (Thermo Fisher Scientific). Gene expression was normalized using the ΔΔct method using the 16S ribosomal RNA (*16S rRNA*) [93] and DNA-dependent RNA polymerase (*rpoB*) mRNA as reference transcripts. Probes, if used, carry a 5′ FAM and a 3′ BHQ2 modification. The following TaqMan PCR primers and probes were used (all 5′→3′; primer forward; primer reverse; probe, if used):
*16S rRNA*: CGGTGAATACGTTCYCGG; GGWTACCTTGTTACGACTT; CTTGTACACACCGCCCGTC*rpoB*: GATGCGTCCCGTATCGTTATC; CTGGTTAGAGCGGGTGTATTT;*ryhB1*: TACGGAGAACCTGAAAGCAC; AATAATACTGGAAGCAATGTGAG;

### 4.7. Statistical Analysis and Data Handling

Statistical analysis and graphical depiction were performed using Graph Pad Prism (Version 9.1.0). Due to a log-normal distribution of bacterial growth the raw CFU/mL data were logarithmized and normalized to the mean of the respective control group of each independent experiment if not mentioned otherwise. After normalization, outliers were identified by Tukey’s boxplot method, whereas an outlier was defined as being over 1.5 times the interquartile range below or above the first and third quartile, respectively [94]. Outliers were excluded from data analysis. All graphs in this paper are depicted with the mean ± the standard deviation (SD). Significant differences, determined by one or two way ANOVA with post-hoc analysis, were corrected for multiple comparisons using the Holm-Sidak method. *p* values are either shown as asterisks (0.01 to 0.05 = *, 0.001 to 0.01 = **, 0.0001 to 0.001 = ***, <0.0001 = ****), or as non-significant (numeric *p*-value). The level of significance was set at *p* = 0.05.

## 5. Conclusions

Ferritin is an acute-phase reactant and as such is considered a pathogen-proof iron storage. We herein demonstrated that *S.*Tm. and *E. coli* are able to mobilize small amounts of iron from ferritin, sufficient to promote bacterial growth. Although siderophore-mediated iron uptake from ferritin predominates, the redundancy of bacterial iron acquisition and our results suggest that reductive uptake via the Feo and SitABCD systems is relevant for bacterial growth, as well. We could also demonstrate that ferritin-bound iron is less available for *S.*Tm. and *E. coli* than free ionic iron. Hence, ferritin does contribute to iron sequestration from Gram-negative bacteria. Furthermore, ferritin and SOD undermine the reductive mechanisms of bacterial iron uptake, thus cooperating in nutritional immunity. Future studies are needed to characterize these interactions on a molecular level in order to identify possible targets for therapeutic interventions.

## Figures and Tables

**Figure 1 ijms-23-13087-f001:**
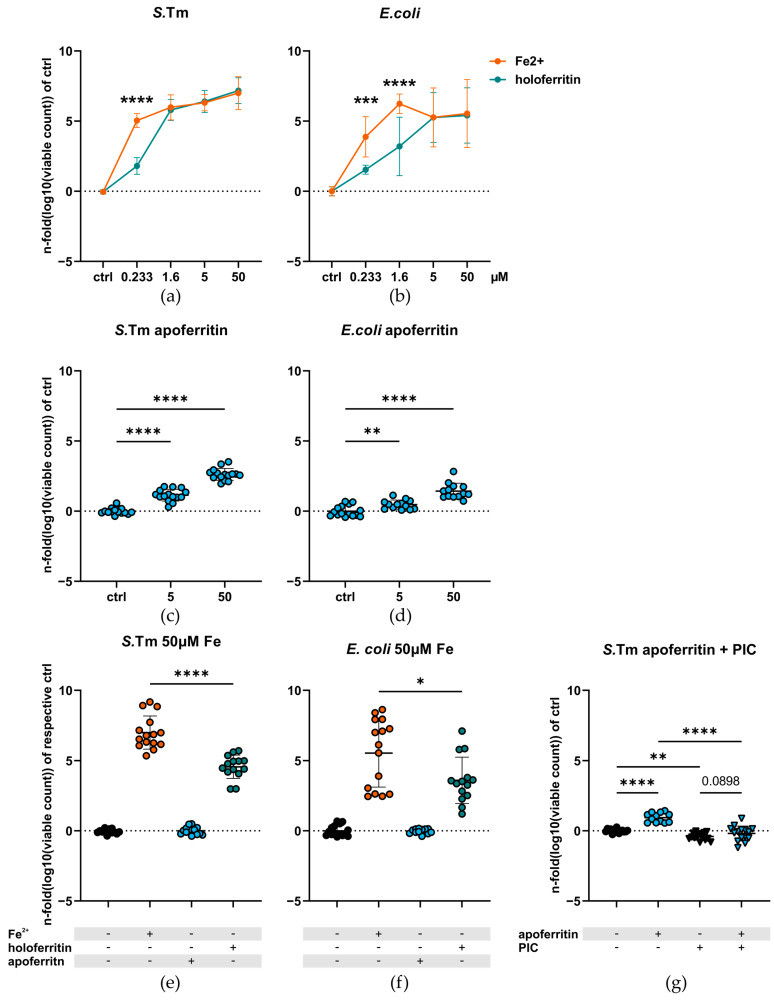
Growth of (**a**) *S.*Tm (N = 15, n = 5) and (**b**) *E. coli* (N = 15, n = 8) upon supplementation of ferrous iron and holoferritin at rising iron concentrations. Utilization of apoferritin by (**c**) *S*.Tm and (**d**) *E. coli* in equal protein concentrations corresponding to holoferritin containing 5 and 50 µM of iron (0.0028 and 0.0282 µM apoferritin, respectively; N = 15, n = 5). The addition of apoferritin resulted in a very small increase in the iron content of the final solution, which we calculated to be 1.121 and 11.221 nM for the low and high apoferritin group, respectively. Growth of (**e**) *S.*Tm (N = 15, n = 5) and (**f**) *E. coli* (N = 15, n = 5) upon the addition of 50 µM iron as ferrous iron or holoferritin normalized to their respective control: For ferrous iron, IMDM alone, whereas for holoferritin, apoferritin in equimolar protein concentration was used for normalization. (**g**) The growth-promoting effect of apoferritin on *S.*Tm upon addition of a PIC. All cultures were grown aerobically in ambient air at 37 °C and were shaken at 200 RPM. Growth data were logarithmized and then normalized to their corresponding control. Graphs are depicted as mean ± SD. (**a**,**b**) Two-way ANOVA or (**c**,**d**) one-way ANOVA corrected for multiple comparisons with the Holm-Sidak post hoc test was used. (**e**–**g**) Student’s t-test was used (*: *p* < 0.05; **: *p* < 0.01; ***: *p* < 0.001, ****: *p* < 0.0001). N: number of biological replicates, n: number of experiments conducted, PIC: protease inhibitor cocktail.

**Figure 2 ijms-23-13087-f002:**
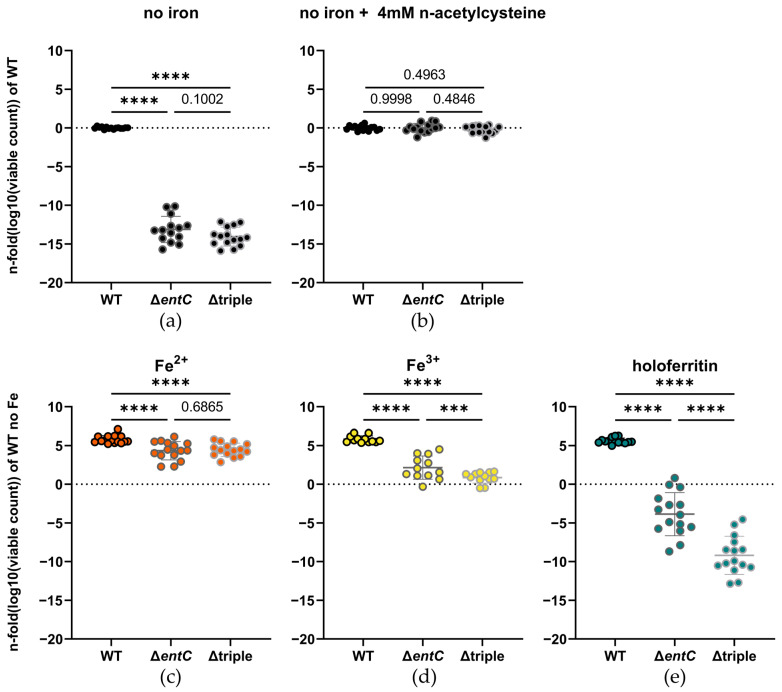
*S.*Tm ATCC 14028 and derivative knockout mutant strains were grown (**a**) without any supplement (N = 15, n = 5) or (**b**) with the addition of 4mM of n-acetylcysteine (N = 15, n = 5). Growth of the same strains with 5 µM (**c**) of ferrous iron (N = 15, n = 5), (**d**) of ferric iron (N = 12, n = 5) or (**e**) of ferritin-bound iron as sole iron source (N = 15, n = 5). Growth data were logarithmized and then normalized to the WT without any iron supplement. Graphs are depicted as mean ± SD. For statistical analysis (**a**–**e**) one-way ANOVA corrected for multiple comparisons with the Holm-Sidak post hoc test was used (***: *p* < 0.001, ****: *p* < 0.0001, numbers indicate a non-significant *p*-value > 0.05). N: number of biological replicates, n: number of experiments conducted, WT: wildtype, Δ*entC*: *S.*Tm enterobactin/salmochelin knockout mutant strain, Δtriple: combined knockout mutant strain of *entC*, *sit* and *feo*.

**Figure 3 ijms-23-13087-f003:**
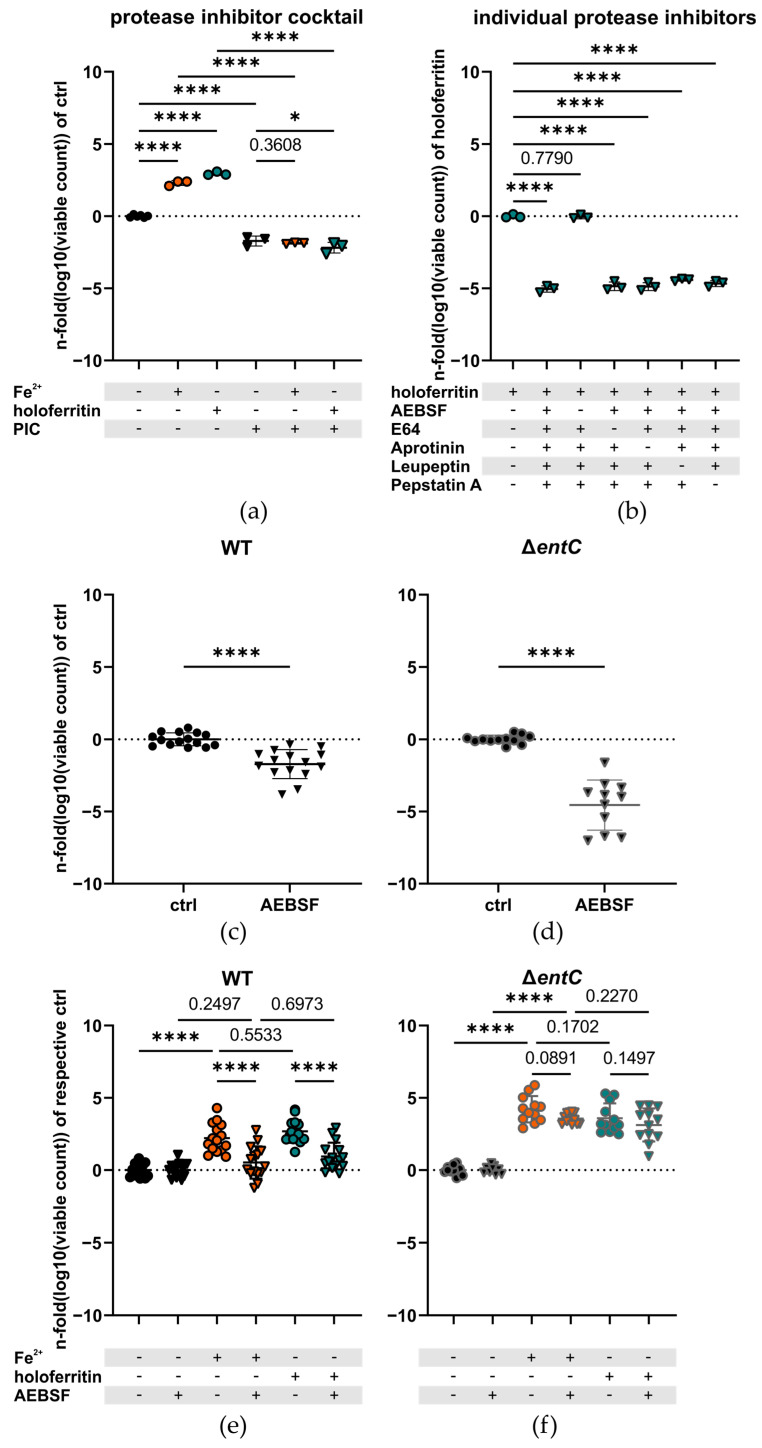
(**a**) Growth of *S.*Tm upon addition of a PIC (N = 3, n = 2). (**b**) Effect of the individual protease inhibitors of the PIC (excluding Bestatin hydrochloride, which was not available due to shortness of supplies, AEBSF: 346.67 µM, E64: 4.67 µM, Aprotinin: 0.27 µM, Leupeptin: 6.67 µM, Pepstatin A: 5 µM), on the growth of *S.*Tm with 5 µM of ferritin-bound iron (N = 3, n = 1). Growth of (**c**) WT *S.*Tm (N = 15, n = 5) and (**d**) the Δ*entC* strain upon addition of AEBSF to the medium (N = 12, n = 4). Growth of (**e**) the WT strain (N = 15, n = 4) and (**f**) the Δ*entC* mutant (N = 12, n = 4) upon addition of ferrous or ferritin bound iron with or without AEBSF (triangles and circles, respectively). (**a**,**b**) were conducted in ambient air and (**c**–**f**) were conducted in 5% O_2_, 5% CO_2_ atmosphere. Growth data were logarithmized and then normalized to (**a**,**c**,**d**) the iron free control, (**b**) supplementation with 5 µM of ferritin-bound iron, or (**e**,**f**) their corresponding controls: no supplement (circles) or AEBSF (triangles), to account for the iron-independent growth inhibition of AEBSF. Graphs are depicted as mean ± SD. For statistical analysis, either Student’s t-test (**c**,**d**) or one-way ANOVA (**e**,**f**) corrected for multiple comparisons, and the Holm-Sidak post hoc test was also used (*: *p* < 0.05; ****: *p* < 0.0001, numbers indicate a non-significant *p*-value > 0.05). N: number of biological replicates, n: number of experiments conducted, PIC: protease inhibitor cocktail, WT: wildtype *S.*Tm, Δ*entC*: *S.*Tm enterobactin/salmochelin knockout mutant strain, AEBSF: 4-(2-Aminoethyl)benzenesulfonyl fluoride.

**Figure 4 ijms-23-13087-f004:**
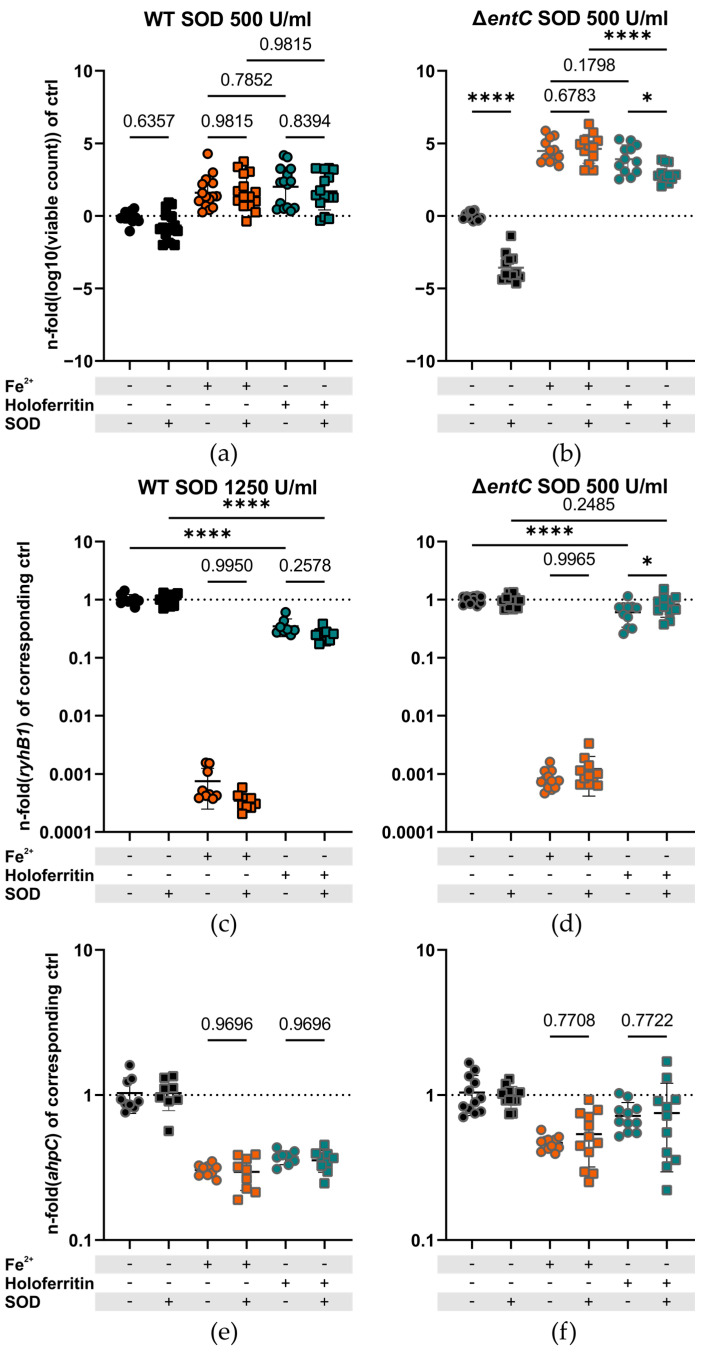
Iron-dependent growth of (**a**) WT *S.*Tm (N = 15, n = 5) and (**b**) the Δ*entC* strain with or without SOD, depicted as squares and circles, respectively (N = 12, n = 4). Fold change in *RyhB1* sncRNA expression normalized to the mean of the corresponding iron-free control of (**c**) WT *S.*Tm (N = 9, n = 3) and (**d**) the Δ*entC* strain upon different iron sources with and without SOD (N = 12, n = 4). Fold change in *ahpC* expression normalized to the mean of the corresponding iron-free control of (**e**) WT *S.*Tm (N = 9, n = 3) and (**f**) the Δ*entC* strain upon different iron sources with and without SOD (N = 12, n = 4). (**c**,**e**) In the WT a higher concentration of SOD was used in order to ensure complete dismutation of O_2_^-^ to assess a possible involvement of the catalytic mechanism with strong iron(III) ligands [57]. Cultures were incubated at 5% O_2_ 5% CO_2_, 37 °C and 200 RPM. Growth data were logarithmized and then normalized to the corresponding control without any supplement. Expression data were normalized to their corresponding controls: no supplement (circles) or SOD (squares). Gene expression was normalized using the ΔΔct method using the *16S* ribosomal RNA (*16S* rRNA) and DNA-dependent RNA polymerase (*rpoB*) mRNA as reference transcripts. Graphs are depicted as mean ± SD. For statistical analysis, one-way ANOVA (**a**–**f**) corrected for multiple comparisons when the Holm-Sidak post hoc test was used (*: *p* < 0.05, ****: *p* < 0.0001, numbers indicate a non-significant *p*-value > 0.05). N: number of biological replicates, n: number of experiments conducted. SOD: superoxide dismutase, WT: wildtype *S.*Tm, Δ*entC*: *S.*Tm enterobactin/salmochelin knockout mutant strain, SOD: superoxide dismutase.

## Data Availability

The raw data supporting the conclusions of this paper will be provided upon request without undue delay.

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
