# Peer review of "Availability of Ferritin-Bound Iron to Enterobacteriaceae"

_ijms, 2022, doi:10.3390/ijms232113087_

Round 1

Reviewer 1 Report

This is an interesting paper examining the contributions of ferritin iron on bacterial growth.  It is evident that iron is available as a result of proteolysis of ferritin or by proteolytic enzymes or by reduction of the ferrihydrite core content by superoxide or possibly other reductants.  

I am a little concerned about some of the statements in the introduction.  in line 87 enterobactin is also important as a radical scavenger [34].  It seems likely that the radical scavanging activity is a consequence of the  hexadentate iron chelation by enterobactin .

I am unclear what point the authors are trying to make by use of apoferritin in figure 1 c and 1d. It would appear to promote slightly more growth in S.Tm than E.Coli which is prevented by PIC Fig 1g.   What specific contribution is apoferritin supposed to be making,  iron release, due the presence of residual iron (not measured), or iron binding due to  ferroxidase activity, or to its intrinsic SOD activity see RSC Adv.,2021, 11,26211  DOI: 10.1039/d1ra03816h.

Additionally I think the way the data are presented in figure1 c-f is confusing.  The absence of a control without added iron makes it seem as if addition of holoferritin is inhibiting uptake when all they are  showing in reality is that   holoferritin is  slightly inferior to free iron as a growth stimulant.  

In figure 4   added exogenous SOD has no   significant effects.  This should be explained. It might be interesting to observe the effect of exogenous H2O2.  

Additionally I find this paper rather confusingly written.. I could do with a bit of editing and clarification.  For example  

Ferritin is approximately 480 kDa in size and it consists of 2 subunits: An H-subunit,   containing a ferroxidase active site, and an L-subunit, which is primarily involved in iron   nucleation within the ferritin cavity and contributes to ferritin stability [20]. Ferritin’s iron lines 62- 64.  I assume that the authors mean that ferritin consists of 2 TYPES of subunit.  There are normally 24 subunits in ferritin consisting of variable proportions of L and H chains.    

Reviewer 2 Report

This is an interesting study that addresses the ability of Salmonella to utilise host ferritin as an iron source.  Unfortunately, horse spleen ferritin was deployed (not human ferritin) and this aspect is not made clear (lost in the Methods). Horse spleen ferritin was shown to stimulate iron-restricted growth, although it was surprising that apoferritn also had an effect, albeit relatively modest. An attempt to inhibit utilisation of ferritin as an iron source through use of protease inhibitors was not, ultimately, successful; an apparent impact of one inhibitor on Fes is well suggested but not proven (this aspect is somewhat of a distraction from the main theme of the manuscript).  SOD was shown to reduce iron-restricted growth of the entC mutant, and very minor effects of SOD were observed for holoferritin dependent growth and ryhB1 expression. No apoferritin controls were included here, nor was the expression of any other Fur dependent or redox stress dependent gene.

Major issues are an apparent failure to provide protease inhibitor cocktail controls (Fig 1) and the interpretation of the NAC results (no alternative approaches were attempted and no alternative interpretations were considered). In addition, some of the effects observed are very modest (Fig 4 – effect of SOD in presence of holoferritin) and the protease inhibitor work does not provide any insight of relevance to the research question. The degradation of ferritin was not directly tested and so remains unclear.

The composition of the text is a little clunky in places and includes rather too many very short paragraphs.  However, it is largely informative and clear.

English, scientific nomenclature and interpretation need improving in places, key scientific details are missing in places; some Figs/legends are a little unclear. 

In summary, the research area is of great interest but this manuscript provides little advance and requires considerable improvement. Too many uncertainties are apparent and the research has not been conducted in a sufficiently complete fashion as yet.

 Specific points

29           gram to Gram (and elsewhere in the manuscript)

59-61     make clear whether it is the thalassemia itself that causes this effect or whether it is the result of transfusion treatment

62           Here, specify human or mammalian ferritin (as not all ferritins contain 2 subunits).

74           replace relevant text with ‘of a complete H and L ferritin gene knockout’ – no itals.  You must make clear which gene (H or L) and if both, indicate this clearly.

77           is SitABCD also present in the E. coli strain that you deployed?

86           replace ‘dissolving’ with ‘enabling its release from …’

107         but does AEBSF prevent release of iron?

152         replace term ‘solvent’ with ‘growth medium’ throughout

                Make it clear in the abstract and results that horse spleen ferritin was used here.

                Did you confirm that the apo horse spleen ferritin used is entirely iron free?

Fig 1       please indicate the growth conditions briefly in the legend – to assist the reader

Fig 1       please make clear want N and n equate to (biological and technical repeats?)

Fig 1       why not add c to a, and d to b (in line graph form)?  Why start the a axis at -5, why not -1?

Fig 1g legend – includes description of results which should be reserved for main body of the text.

Fig 1g     unfortunately, this fig fails to indicate the effect of PIC alone.  Please include these data. (I see that PIC controls are included later [in Fig 3] – this is too late, and really if these controls had been used in Fig 1 then the use of PIC later on in the manuscript would not have been considered worthwhile).

Fig 1g fails to show the impact of PIC on the growth promoting properties of apoferritn (you need to compare with the growth medium only control), and it is a pity the PIC was not also used in the holoferritin growth promotion studies

182         Δtriple strains’ is incorrect use of genetic nomenclature.  Also, the delta symbol should not be italicised

184         n-acetylcysteine is known as a metal chelator, and thus at 4 mM may well have a major impact on iron availability.  You really need to prove that this compound is working as an antioxidant and not an iron chelator

187         ‘This further corroborates the impact of iron in ROS defense’.  This statement makes little sense to me – the iron uptake mutations would be expected to have little impact/relevance under iron supplemented conditions (the genes would not normally be induced). Further, you do not explain how the results in Fig 2c/d corroborates your asertation – no argument is provided!

192-4     poorly worded – please improve

197-8     obscure as composed – what is meant by ‘other factors’?

199         ‘free iron’ is not the correct description here – these results indicate that ferritin-iron is available.  Also, you did not include the apoferritin control with the mutants, this is a pity

202         there is nothing really to ‘interpret’ here – I think you mean that you ‘hypothesise’ or ‘suggest’ that O2- formation is the cause??

216         is Salmonella known to release extracellular proteases?  If so, please make this clear and provide some detail.  If not, again ensure that this detail is made clear.

220         the non availability of Bestatin hydrochloride should be moved to the corresponding Fig legend.

222-3 (and Fig 3a)            a PIC only control (without any iron addition) is included here. This is a key control and must be included.

226-7     any impact of ROS has not been clearly proven – this statement must be accordingly adjusted

Fig 3ef   the normalisation statement is unclear – please provide a more precise statement here that is entirely clear.  Normalisation must be to the same comparator within each graph (it seems that different comparators are used in these graphs!) or else the results will appear misleading to the reader.

234        WT should not be italicised anywhere in the manuscript!

Fig 3       make absolutely clear what the growth medium employed was and what the level of added iron (and inhibitors) was in each case

Fig 3       AEBSF reduces growth of the entC mutant in Fig 3b and d, but not in Fig 3f.  The reason for this is unclear from the manuscript as currently composed!

Fig 3       the change in growth conditions (oxygen levels) is not indicated in the legend!

232-7     the effect of AEBSF is not well established (see above) – this is a critical issue that must be addressed. 
Here, can you indicate the specificity of AEBSF (a serine protease inhibitor) and therefore the likelihood that it would inhibit Fes?  Did you utilise any other Ser protease inhibitors (e.g. PMSF)?

232-7     I agree with this interpretation, but should you not also consider the salmochelin specific esterases (IroD and E)?

2.4          I can find no information on the source or type of SOD employed.  Was it an Fe-SOD?  Full details are required and an indication of the amount of SOD-associated metal added to each experiment should be included.

Fig 4       Indicate the growth conditions clearly, including the O2 regime.

261         Fig S1 – but what was the result and conclusion!

263-4     Please provide the fold differences in growth and the exact P values

265-6     this is a confusing statement – how can ‘restricted growth’ be too small to determine by growth measurements?

267         RhyB/rhyB – to RyhB/ryhB throughout – I find that this is a surprisingly common error in manuscripts!

270-       The expression data showed no difference in the expression of ryhB1 in the WT strain in response to SOD (Figure 4c), while a significant upregulation of ryhB1 expression was detected in the ΔentC mutant when SOD was added in the presence of holoferritin (Fig 4d)’.  
                This statement should be improved as indicated above. Please indicate the exact difference and direction of the expression difference, and provide the corresponding P value – the difference looks very slight (not very convincing)

270-        the effect of SOD on expression of ryhB1 (Fig 4d) is modest and the effect really needs to be proven more thoroughly using additional targets.  Also, Fur regulation is impacted by redox stress (superoxide and SoxRS) and thus the effect observed may be redox stress rather than iron related

Fig 3c      this fig uses a much higher level of SOD and so cannot be compared to 3d!  Please explain this difference

273-        this is not an ‘upregulation’!  This is a failure to induce a down regulation (as was seen for ferrous iron).  Please reword this statement accordingly.

309-11   This is entirely speculation and has not been explored here.

328         what is ‘relative growth’?            

334-43   this section makes little sense to me since you propose that Fes is the key enzyme subject to inhibition and presumably any protease of relevance would need to be secreted?

344-9     This statement is not required/useful and leads nowhere useful

362-5     I have been unable to find the results you elude to in this statement

386         but levels are very low in the serum and it contains v little iron

460         how does the cell counter determine ‘viability’?  Please make this clear. 

460         Note, CFU is not necessarily the same is viable cell counts. CFU determination would normally be expected to involve the generation and counting of colonies.  Seems that you did not determine CFU, but instead measured total cells (or total viable cells). Please adjust your manuscript accordingly throughout to ensure that you clearly apply the correct terminology.

In general, the Discussion rambles somewhat and would benefit form a reduced content with a clearer focus on the results obtained

                Did you determine the concentration of iron in IMDM?

entC KO would also prevent salmochelin production – you must make this clear in your manuscript

529-       I see no evidence for a role for proteases – this conclusion appears false. In general, the Conclusion statement appears poorly considered and should be re written to ensure if accurately reflects the findings of the manuscript

Reviewer 3 Report

In the paragraph 4.4: "Specific experimental condition" you assessed the effect of ROS on the mutant strains via the addition of NAC. Haven't  you  also thought to experimentally determine the presence or absence/decrease of ROS? By flow cytometric assay?

In the paragraph 4.5: "Quantification of iron released from ferritin". The micro molarities of the iron involved are very low. Have all the solutions and buffers used in the experiments been pre-treated with Chelex to minimize iron contamination?

Round 2

Reviewer 2 Report

The authors have responded well, on the whole, to this reviewer’s concerns and have consequently modified the manuscript considerably

However, a few issues are still apparent - these should be easy to address

1.       Do not italicise ‘H-ferritin’ gene or ‘L-ferritin’ gene/allele!  Only ‘FTH1’ or ‘FTL’ should be italicised

2.       Please include the measured levels of iron in IMDM within your manuscript

3.       It would be a good idea to specify the resulting (very minor) contribution of apoferritin to the total iron content of the medium (in your results – figure legend perhaps)

4.       Fig 1.    d – ‘coli’ in itals; Fig1c/d – units missing for x axis; best place letters (indicating the subpanels) directly in front of the subpanel titles

5.       236-7:  ‘further corroborates the role of iron in enzymes for ROS defense’.  I entirely disagree with this statement – you would need evidence that the effect you see is redox stress related and not caused by some other effect.  You could say ‘is consistent with’ in place of ‘further corroborate’

6.       248-9: ‘This indicates that not only the oxidation state affects the availability of ferritin-bound iron to the knockout strains’ – but the effect of oxidation state of ferritin iron was not explored!  If this effect is related to work of others, then include the reference in support of your text directly after the corresponding text

7.       251-2: ‘indicating that free iron is present in the medium’.  Ferritin iron is not generally considered as ‘free iron’.  Better state ‘indicating that free iron, released from ferritin, is available in the medium’. This provides clarity I think

8.       255: ‘suggest’ or ‘may be’ – not both

9.       254: ‘found it to be 0.147’. Here, you need to indicate the difference in iron levels measured between the medium with and without holoferritin, and to indicate how the holoferritin molecules were removed (but see below)

10.   253: ‘into the medium (following filtration)’

11.   ‘off ferritin’ – change to ‘from ferritin’ throughout

12.   607: ‘this finding demonstrates’ – change to ‘this finding suggests’

13.   ‘Lon protease’ not ‘lon protease’

14.   871: ‘   I see no good evidence that you have proven that a reductive mechanism is involved in holoferritin iron acquisition.  In Fig 7, the growth suppression of the entC strain by SOD is freater without ferritin (b) than with ferritin (d).

15.   875-6: I don’t agree with this statement – it doesn’t seem well supported by your results.

16.   876-8: it is difficult to follow the meaning of this statement

17.   Conclusion: this statement is very poor and should be greatly improved

18.   Fig3C – higher levels of SOD used.  Please provide the explanation in the Fig legend
